# Are the healthcare providers willing and able to respond to disasters: An assessment of tertiary health care system of Khyber Pakhtunkhwa

**Muhammad Zeeshan Haroon** [1], **Inayat Hussain Thaver**[2], **Muhammad Imran Marwat**[3]*

**1** Department of Community Medicine Ayub Medical College, Abbottabad, Pakistan, **2** Department of Community Health Sciences, Bahria University Medical and Dental College, Karachi, Pakistan, **3** Department of Community Medicine and Public Health, Khyber Medical College, Peshawar, Pakistan

* imranmarwat_30@kmc.edu.pk

**Data Availability Statement:** The data is now available at following URL on OSF https://osf.io/numeh/?show=revision&view_only=18026d1679104d709462a3ab4f54b6ed.

## Abstract

For the tertiary health care system to provide adequate care during disasters, willing and able healthcare providers must be available to respond to the abnormal surge of the patients. Health care professionals (HCPs) constantly face a dilemma because of their profession to either respond to disasters or protect themselves. This study was conducted to assess the willingness and ability of HCPs working in the tertiary healthcare system of Khyber Pakhtunkhwa to respond to disasters. This cross-sectional survey was conducted in all the 8 tertiary care hospitals of the Khyber Pakhtunkhwa province of Pakistan. For different disaster scenarios, between 6% and 47% of HCP indicated their unwillingness, and between 3% & 41% of HCPs indicated that they were unable to respond to the given disaster scenarios. HCPs with childcare obligation indicated significantly lower willingness (p<0.05) to respond to earthquakes, MCIs, and an outbreak of Influenza, and SARS. Male HCPs showed a significantly (p<0.05) higher willingness to respond to earthquakes, MCIs, and an outbreak of Influenza as compared to their female counterparts. The overall ability indicated by HCPs for various disaster scenarios ranged between 54.1% [95% CI 0.503,0.578] for responding to victims of nuclear war and 96.4% [95% CI 0.947,0.976] for responding to conventional war. The HCPs who indicated childcare obligation showed a significantly lower ability (p<0.05) to respond to environmental disaster, influenza outbreak, and responding to victims of nuclear war. Female HCPs indicated significantly higher ability (p<0.05) as compared to their male counterparts. This survey provides an opportunity for the tertiary healthcare system to build on the findings and develop disaster mitigation plans to address the barriers to improving the HCPs' availability during disasters.

## Introduction

The healthcare system of any country is a major asset during any disaster or emergency response. During a successful disaster response, the health care system requires two types of

**Funding:** The authors received no specific funding for this work.

**Competing interests:** The authors have declared that no competing interests exist.

resources: a. Hard resources (medicine, equipment, beds, supplies, HR, etc.) b. Soft resources (skill, knowledge, expertise, willingness, etc.) [1]. In developed countries, the health system may have both adequate hard and soft resources, but in resource constrained countries like Pakistan, where the tertiary healthcare system is stretched past its capacities day and day out [2], there is a very strong chance that the system might not only have a shortage of hard resources but due to lack of training, resulting from lack of resources and commitment, the soft elements may also be deficient. The healthcare system with these deficiencies can and will be overwhelmed by the disasters and an adequate response may not be possible [3]. For the tertiary health care system to provide adequate care during a disaster and public health emergencies, adequate willing healthcare staff needs to be present and they must be appropriately trained to be able to respond to the abnormal surge of the patients and provide the necessary care.

Recently, the world has seen an unprecedented number of disasters and public health emergencies, such as terrorism events, abnormal environmental phenomena like hurricanes, tornados, floods, and diseases outbreak and epidemics. These disasters have increased fourfold over the last two decades. A study has shown that natural hazards globally have increased from 120 per year in the 80s to around 500 in the 21st century [4]. In the last two decades, Pakistan has seen a large number of disasters whether natural hazards or, more importantly, and frequently, manmade. During this period, terrorism has cost Pakistan around US$ 126.79 billion in direct and indirect costs [5]. The province of Khyber Pakhtunkhwa (KP) has especially been affected by manmade disasters in the form of suicide attacks, mass shootings in addition to natural hazards, and massive disease outbreaks like Dengue, Crimean Congo Hemorrhagic Fever, Cholera, Malaria, Measles, and other vaccine preventable diseases. These events have put significant strain on the tertiary health care system of KP and the available resources are stretched thin. In KP, there are 2451 primary healthcare facilities, 269 secondary and 8 tertiary care hospitals. Approximately 10325 health providers which include doctors, nurses, and paramedics (5489, 2517, 2319 respectively) are working in tertiary care hospitals across Khyber Pakhtunkhwa [6]. The health-care system of KP faces several issues such as scarce resources, untrained as well as insufficient manpower, inequity, gender inequality, and structural mismanagement. In addition to the shortage of HRH, certain system practices and situations such as repeated disasters, either manmade or natural, have further intensified the strain on the already inadequate health workforce working in the tertiary health care system of the province.

The driving force behind any health system is the health workforce. This driving force does not only depend upon availability but additionally also depends upon the motivation and competence of the health workforce, equitable distribution, accessibility of the population to the health workforce, and support from the system [7]. Human resource role and importance cannot be denied in routine service delivery or during disaster response.

Health care professionals (HCPs) constantly face a dilemma because of their profession to either respond to disaster or protect themselves. The HCPs are part of the disaster affected community and also have a social commitment to care for the community in which they live and serve. Feeling this obligation, the HCPs are pressured into performing the duty even if knowing that providing a response can put them at risk and may kill them. This was evident in the 2003 SARS outbreak in which 21% of the health workers while providing care got infected with SARS and even transmitted this infection to their families with two of them dying later [8, 9]. WHO estimates that HCPs are 21–32 times more prone to get infected as compared to the general population [10].

An extensive review of the literature did not reveal a single study showing either 100% willingness or availability of HCPs to respond to a disaster or public health emergency [11–13].

Studies have shown the willingness and ability of HCPs to respond to disasters between 18%-90% [14–16]. Published literature suggests that willingness and ability are influenced by the type of disaster or threat. In addition to the type of disaster, they are dependent upon the support from the health system that includes the provision of personal protective equipment, appropriate vaccine, and other supportive measures [17].

This study aims to assess the willingness and ability of HCPs working in tertiary care hospitals during various possible disaster scenarios and identify factors that affect their willingness and ability to respond thereby, providing evidence for the health managers to plan for adequate human resources, factoring in possible absenteeism due to lack of either willingness and or the ability of the HCPs, to respond to various disaster scenarios.

## Methodology

This was a cross-sectional survey conducted in all the 8 tertiary care hospitals of the Khyber Pakhtunkhwa province of Pakistan. This study was conducted between February 2019 and August 2019. Ethical approvals were obtained from the ethics review committees of Ayub Medical College Abbottabad, Khyber Medical College and post graduate ethics committee of Lady Reading Hospital Peshawar under ethical approval letters No 396/ADR/KMC and No.04/R&T/BKMC. Furthermore, data collection permissions were obtained individually from Medical/Hospital Director of each tertiary care hospital..

The willingness and ability have been measured in the past by qualitative studies [18, 19], quantitative studies [16, 20, 21], or by mixed design studies [17]. For this study, a perception based approach which is a quantitative approach was used to assess the extent to which the HCP are able and willing to respond to disasters. This approach is favored owing to its ability to quantify behaviors which helps in understanding the phenomenon and allows comparison with other studies. Furthermore, it is near impossible to experience every type of disaster and this approach by generating multiple disaster scenarios can explore the willingness and ability of HCPs without them necessarily experiencing all the disaster scenarios.

### Survey tool

The basis for the tool used in this survey was the Disaster survey tool designed and used by Qureshi et al. [21]. This is a valid tool that has been used in multiple studies with a large sample size [15, 21, 22]. The questionnaire included 19 major items with multiple sub-items. The questionnaire, which included eight disaster scenarios both natural hazards and manmade, was designed to assess the willingness and ability of participants to respond during various scenarios of disaster and public health emergencies. Certain scenarios were added to the original questionnaire based on already experienced disasters. These scenarios are given in Table 1. Content validation of the modified tool was carried out using the content validity index (CVI).

The questionnaire in addition to containing demographic data and variables on the willingness and ability includes the definition of Willingness and Ability. Willingness was defined as "voluntary intention to report to duty" and Ability was defined as "physical or mental health, personal obligations, other obligations, competence or circumstances that might hamper the ability to report to duty in his or her usual capacity" [23].

### Sample size

The sample size calculated as 758. Assumption of 20 percent was taken as potential non responders owing to various studies showing high non response rate.

The survey enrolled 758 HCPs using WHO sample size calculator with assumption of willingness in HCP to respond to disaster taken as 18% [11] with 95% Confidence interval and 4%

**Table 1. Disaster scenarios.**

| Type of Disaster | Type of Event | Scenario |
|---|---|---|
| Natural Hazards | Environmental | 450 mm or more of heavy monsoon rains in 24 hours |
| | Earthquake | Magnitude 7 earthquake with multiple after shocks |
| | Outbreak of Treatable infection | Outbreak of influenza |
| | Outbreak of Untreatable infections | Outbreak of SARS |
| Man Made | Mass Casualty Incident | Explosion with multiple fatalities and injuries brought to the hospital |
| | Chemical | Multiple victims of chemical injuries brought to the hospital due to corrosive chemical spillage |
| | Conventional War | Multiple victims brought to the hospital due to injuries sustained in a conventional war |
| | Nuclear War | Multiple victims brought to hospital with nuclear radiation injuries caused by a nuclear bomb blast/fall out. |

of relative precision. The enrolled HCPs included doctors, nurses, and paramedical staff of all the tertiary care hospitals of KP province. Approximately 10325 health providers which include doctors, nurses, and paramedics (5489, 2517, 2319 respectively) are working in tertiary care hospitals across KP. The participants were selected through stratified random sampling. The sample of 758 participants was stratified in the 8 tertiary care hospitals of Khyber Pakhtunkhwa. The sample size was then proportionately allocated among the doctors, nurses, and paramedics based on their cadre's strength in that hospital. The doctors, nurses, and paramedics were then randomly selected through payroll and HR database. Self administered questionnaire was distributed amongst the selected participants.

The survey was anonymous and the participants were assured of anonymity and confidentiality. The consent was implied when the HCPs returned the filled questionnaire which included a consent statement. The survey was piloted in one of the major tertiary care hospitals of the KP province.

The collected data were analyzed using SPSS v.27. Proportions and 95% confidence interval were calculated for willingness and ability of the HCPs to respond against various disaster scenarios. Chi Square test and Cramer V statistical tests were applied to assess the association and strength of association between the variables.

## Results

The overall response rate of the survey was 91.68% (695/758). The mean age of the surveyed HCPs was 35.01 ± 7.56 years. More than half 407 (58.6%) of the HCPs were between 25–35 years. The majority 369 (52.9%) of the responding HCP were male participants. The gender distribution was quite diverse in the three cadres of HCPs, male constituted 57.9% of the responding doctors. The overwhelming majority of the responding nurses 157 (92.4%) were female. Inversely, 142 (90.4%) of the paramedics were male.

The mean working experience of all the surveyed HCP was 9.69 ± 7.71 years. The mean working experience of doctors working in the teaching hospital was 6.62 ±7.03 years. Whereas nurses had 13.17± 7.14 years of experience, and a mean of 11.12± 6.77 years of experience was recorded for paramedics.

When HCPs were asked if they were willing to additional shifts while responding to a disaster, the majority 580 (83.5%) [95% CI 0.804,0.861] responded positively to the suggestion. This

trend was observed across all three cadres of HCPs. Of the total 580 HCPs who responded positively to the additional shifts, most of the HCPs 288 (49.65%)[95% CI 0.455,0.538] suggested that they were willing to perform their duty in any shift that necessitates their presence.

The survey showed that 148 (21.3%) of the HCPs had the obligation of childcare. Similarly, 163 (23.5%) of HCPs also reported elderly care obligations. Nurses 93 (54.7%) reported the highest number of childcare obligations. Whereas, the doctors 94 (25.5%) reported the highest number of elderly care obligation.

The highest level of concern by the HCPs 369 (53.1%) [95% CI 0.493,0.568] was shown in the scenario of nuclear war followed by 362 (52.08%) [95% CI 0.483,0.558] of HCPs showing high concern in a scenario of responding to an outbreak of SARS. This trend was observed across doctors, nurses, and paramedics. The lowest level of concern by the HCPs was observed in the scenario of Mass Causality Incidents (MCIs) in which 347 (49.92%) [95% CI 0.461,0.537] of HCPs showed low concern.

Overall willingness to respond to the given scenarios as shown in Table 2 ranged between 93.85% [0.92,0.956] for conventional war scenario and 51.7% [0.479,0.554] for responding to patients presenting with injuries related to nuclear war/fallout. The participants, except for responding to the outbreak of untreatable infection (SARS) and responding to victims of nuclear war, showed more than 70% willingness for the rest of the six disaster scenarios.

When these responses were stratified based on the professional cadres of the participants, the survey results indicated that doctors showed the highest willingness in responding to an earthquake, MCIs, and responding to nuclear war. Similarly, paramedics also showed the highest willingness in three scenarios that included conventional war, an outbreak of Influenza, and responding to environmental disasters. Nurses indicated the highest willingness in the scenarios of responding to chemical spillage and the outbreak of SARS. Interestingly, the surveyed HCPs were more willing to respond to a manmade disaster as compared to a natural hazard. The mean willingness percentage for natural hazards was 72.45%±15.9% [95% CI 0.712,0.736] compared to 75.21%±17.95% [95% CI 0.738,0.765] for manmade disasters.

The majority of HCPs 365 (51.2%) indicated that the reason for them not being willing to respond to given disaster scenarios was a concern for their family. This was reflected across all the three working cadres of HCPs as shown in Table 3.

The survey results, as indicated in Table 4, show that there was a statistically significant correlation between the willingness of HCPs and childcare obligation. The HCPs with childcare obligation showed significantly lower willingness as compared to HCPs with no childcare obligations when responding to an earthquake (p = 0.00 and Cramer V = 0.36), responding to MCIs (p = 0.00 Cramer V = 0.309), responding to an outbreak of Influenza (p = 0.001 and Cramer V = 0.144) and responding to an outbreak of SARS (p = 0.029 and Cramer V = 0.101). If the willingness was stratified based on cadres, a significant difference in willingness with child care was observed in nurses (p < 0.05, Cramer V 0.48) and paramedics ((p < 0.05, Cramer V 0.46). No significant difference was observed among doctors regarding childcare obligation and willingness to respond to the earthquake.

When willingness to respond to different disasters was correlated with gender, it was found that apart from responding to nuclear war patients, where female HCPs (51.7%) were more likely to respond as compared to their male counterparts (50.3%), the rest of the scenarios presented a uniform willingness picture where male HCPs showed more willingness to respond to a disaster as compared to female HCPs. Male HCPs showed significantly more willingness to respond to MCIs, earthquakes and outbreak of influenza as shown in Table 5. All three cadres of HCPs showed similar trends.

When age groups of HCPs were correlated with the various disaster scenarios, no significant correlation was found between them and this was the case for doctors, nurses, and

**Table 2. Willingness of healthcare providers.**

| Scenario | Willing<br>n(%) [95%CI] | Not Willing<br>n(%) [95%CI] | Not Sure<br>n(%) [95%CI] |
|---|---|---|---|
| **Environmental Disaster** | **623 (89.6) [0.871,0.918]** | **30 (4.3) [0.029,0.061]** | **42 (6.0) [0.043,0.08]** |
| • Doctors | 331 (89.9) [0.869,0.928] | 14 (3.8) [0.018,0.058] | 23 (6.2) [0.038,0.087] |
| • Nurses | 147 (86.5) [0.804,0.912] | 10 (5.9) [0.028,0.105] | 13 (7.6) [0.041,0.127] |
| • Paramedics | 145 (92.4) [0.870,0.960] | 6 (3.8) [0.014,0.081] | 6 (3.8) [0.014,0.081] |
| **Earthquake** | **507 (72.9) [0.696,0.763]** | **76 (10.9) [0.087,0.135]** | **112 (16.1) [0.135,0.190]** |
| • Doctors | 299 (81.2) [0.769,0.851] | 23 (6.2) [0.040,0.092] | 46 (12.5) [0.093,0.163] |
| • Nurses | 88 (51.8) [0.439,0.595] | 33 (19.4) [0.137,0.261] | 49 (28.8) [0.221,0.362] |
| • Paramedics | 120 (76.4) [0.690,0.828] | 20 (12.7) [0.079,0.189] | 17 (10.8) [0.064,0.167] |
| **Outbreak of Influenza** | **522 (75.1) [0.719,0.783]** | **82 (11.8) [0.094,0.144]** | **91 (13.1) [0.106,0.158]** |
| • Doctors | 266 (72.3) [0.674,0.768] | 49 (13.3) [0.100,0.172] | 53(14.4) [0.109,0.184] |
| • Nurses | 123 (72.4) [0.649,0.789] | 24 (14.1) [0.092,0.202] | 23 (13.5) [0.087,0.196] |
| • Paramedics | 133 (84.7) [0.781,0.899] | 9 (5.7) [0.026,0.106] | 15 (9.6) [0.054,0.152] |
| **Outbreak of SARS** | **363 (52.2) [0.485,0.559]** | **135 (19.4) [0.165,0.225]** | **197 (28.3) [0.250,0.318]** |
| • Doctors | 193 (52.4) [0.472,0.576] | 67 (18.2) [0.144,0.225] | 108 (29.3) [0.247,0.342] |
| • Nurses | 90 (52.9) [0.451,0.606] | 37 (21.8) [0.158,0.287] | 43 (25.3) [0.189,0.325] |
| • Paramedics | 80 (51.0) [0.460,0.621] | 31 (19.7) [0.138,0.268] | 46 (29.3) [0.223,0.370] |
| **Mass Casualty Incident** | **576 (82.9) [0.801,0.857]** | **43 (6.2) [0.045,0.082]** | **76 (10.9) [0.087,0.135]** |
| • Doctors | 317 (86.1) [0.821,0.895] | 18 (4.9) [0.029,0.076] | 33 (9.0) [0.062,0.123] |
| • Nurses | 124 (72.9) [0.656,0.794] | 17 (10.0) [0.059,0.155] | 19 (17.1) [0.068,0.169] |
| • Paramedics | 135 (86.0) [0.795,0.910] | 8 (5.1) [0.022,0.097] | 14 (8.1) [0.049,0.145] |
| **Chemical Spillage** | **503 (72.4) [0.69,0.757]** | **73 (10.5) [0.083],0.130]** | **119 (17.1) [0.143,0.201]** |
| • Doctors | 264 (71.7) [0.668,0.762] | 40 (10.9) [0.078,0.145] | 64 (17.4) [0.136,0.216] |
| • Nurses | 133 (78.2) [0.712,0.841] | 12 (7.1) [0.037,0.120] | 25 (14.7) [0.097,0.209] |
| • Paramedics | 106 (67.5) [0.595,0.747] | 21 (13.4) [0.084,0.197] | 30 (19.1) [0.138,0.268] |
| **Conventional War** | **652 (93.85) [0.92,0.956]** | **25 (3.6) [0.023,0.052]** | **18 (2.6) [0.015,0.040]** |
| • Doctors | 346 (94.0) [0.910,0.962] | 12 (3.3) [0.017,0.056] | 10 (2.7) [0.013,0.049] |
| • Nurses | 155 (91.2) [0.858,0.949] | 9 (5.3) [0.024,0.098] | 6 (3.5) [0.013,0.075] |
| • Paramedics | 151 (96.2) [0.918,0.985] | 4 (2.5) [0.007,0.0639] | 2 (1.3) [0.001,0.045] |
| **Nuclear War** | **359 (51.7) [0.479,0.554]** | **117 (16.8) [0.141,0.198]** | **219 (31.5) [0.280,0.351]** |
| • Doctors | 193 (52.4) [0.472,0.576] | 65 (17.7) [0.139,0.219] | 110 (29.9) [0.252,0.348] |
| • Nurses | 85 (50.0) [0.422,0.577] | 30 (17.6) [0.122,0.242] | 55 (32.4) [0.253,0.399] |
| • Paramedics | 81 (51.6) [0.434,0.596] | 22 (14.0) [0.089,0.204] | 54 (34.4) [0.270,0.423] |

**SARS:** Severe Acute Respiratory Syndrome

paramedics alike. Analysis of willingness based on years of experience, as shown in Table 6, revealed that except for responding to earthquakes and MCIs, the HCPs generally having more than 10 years of experience were more willing as compared to the HCPs with lesser experience. A significant difference ($p = 0.028$ Cramer V = 0.125) in willingness was only observed

**Table 3. Barriers to willingness to respond to disasters.**

| Health care providers | Concern for Family n(%) [95%CI] | Concern for Self n(%) [95%CI] | Medical Condition n(%) [95%CI] | Other Obligation n(%) [95%CI] |
|---|---|---|---|---|
| Doctors | 182 (49.5) [0.442,0.546] | 102 (27.7) [0.232,0.3259] | 32 (8.7) [0.060,0.120] | 52 (14.1) [0.107,0.181] |
| Nurses | 92 (54.1) [0.463,0.617] | 49 (28.8) [0.221,0.362] | 2 (1.2) [0.001,0.041] | 27 (15.9) [0.107,0.222] |
| Paramedics | 82 (52.2) [0.441,0.602] | 46 (29.3) [0.223,0.370] | 1 (0.6) [0.0002,0.035] | 28 (17.8) [0.121,0.247] |
| **Total** | **356 (51.2) [0.47440.5500]** | **197 (28.3) [0.2502,0.3185]** | **35 (5) [0.035,0.069]** | **107 (15.4) [0.127,0.183]** |

**Table 4. Childcare obligation and willingness to respond to disasters.**

| Scenario | Willing n(%) [95%CI] | Not Willing n(%) [95%CI] | Not Sure n(%) [95%CI] | p | Cramer V |
|---|---|---|---|---|---|
| **Environmental Disaster** | | | | | |
| Obligation of Childcare | | | | | |
| • Yes | 130 (87.8) [0.814,0.926] | 6 (4.1) [0.015,0.086] | 12 (8.1) [0.042,0.137] | 0.49 | 0.045 |
| • No | 493 (90.1) [0.873,0.925] | 24 (4.4) [0.028,0.064] | 30 (5.5) [0.037,0.077] | | |
| **Earthquake** | | | | | |
| Obligation of Childcare | | | | | |
| • Yes | 62 (41.9) [0.338,0.502] | 35 (23.6) [0.170,0.313] | 51 (34.5) [0.268,0.427] | 0.00 | 0.364 |
| • No | 445 (81.4) [0.778,0.845] | 41 (7.5) [0.054,0.100] | 61 (11.2) [0.086,0.140] | | |
| **Outbreak of Influenza** | | | | | |
| Obligation of Childcare | | | | | |
| • Yes | 95 (64.2) [0.559,0.718] | 21 (14.2) [0.090,0.208] | 32 (21.6) [0.152,0.291] | 0.001 | 0.144 |
| • No | 427 (78.1)[0.743,0.814] | 61 (11.2) [0.086,0.140] | 59 (10.8) [0.083,0.136] | | |
| **Outbreak of SARS** | | | | | |
| Obligation of Childcare | | | | | |
| • Yes | 63 (42.6) [0.344,0.509] | 34 (23.0) [0.164,0.306] | 51 (34.5) [0.268,0.427] | 0.029 | 0.101 |
| • No | 300 (54.8) [0.505,0.590] | 101 (18.5) [0.153,0.219] | 146 (26.7) [0.230,0.306] | | |
| **Mass Casualty Incident** | | | | | |
| Obligation of Childcare | | | | | |
| • Yes | 90 (60.8) [0.524,0.687] | 24 (16.2) [0.106,0.231] | 34 (23.0) [0.164,0.306] | 0.000 | 0.309 |
| • No | 486 (88.8) [0.859,0.913] | 19 (3.5) [0.02100.053] | 42 (7.7) [0.055,0.102] | | |
| **Chemical Spillage** | | | | | |
| Obligation of Childcare | | | | | |
| • Yes | 115 (77.7) [0.701,0.841] | 10 (6.8) [0.032,0.120] | 23 (15.5) [0.101,0.224] | 0.172 | 0.071 |
| • No | 388 (70.9) [0.669,0.747] | 63 (11.5) [0.089,0.144] | 96 (17.6) [0.144,0.210] | | |
| **Conventional War** | | | | | |
| Obligation of Childcare | | | | | |
| • Yes | 136 (91.9) [0.862,0.957] | 7 (4.7) [0.019,0.095] | 5 (3.4) [0.011,0.077] | 0.55 | 0.041 |
| • No | 516 (94.3) [0.920,0.961] | 18 (3.3) [0.019,0.051] | 13 (2.4) [0.012,0.040] | | |
| **Nuclear War** | | | | | |
| Obligation of Childcare | | | | | |
| • Yes | 65 (43.9) [0.357,0.523] | 28 (18.9) [0.129,0.261] | 55 (37.2) [0.293,0.454] | 0.10 | 0.081 |
| • No | 294 (53.7) [0.494,0.579] | 89 (16.3) [0.132,0.196] | 164 (30.0) [0.261,0.340] | | |

in the scenario of responding to the outbreak of Influenza which was only evident among the doctors.

Similar to the willingness, the ability to respond to various disaster scenarios also showed dependency upon the type of disaster. The overall ability indicated by HCPs for various disaster scenarios ranged between 54.1% [95% CI 0.503,0.578] for responding to nuclear war victims and 96.4% [95% CI 0.947,0.976] for responding to conventional war. The same trend was observed within the groups of doctors, nurses, and paramedics as shown in Table 6.

The survey indicated, as shown in Table 7, that apart from the ability to respond to MCIs, the HCPs who had the obligation of child care indicated a lower ability to respond to the rest of the seven disaster scenarios. The HCPs who indicated childcare obligation showed a significantly lower ability to respond to environmental disasters (p = 0.008 Cramer V = 0.118), influenza outbreak (p-value = 0.046 with Cramer V = 0.094) and responding to victims of nuclear war (p-value = 0.000, Cramer V = 0.152) as compared to HCPs with no childcare obligation.

**Table 5. Gender and willingness to respond to disasters.**

| Scenario | Willing n(%) [95%CI] | Not Willing n(%) [95%CI] | Not Sure n(%) [95%CI] | *p* | Cramer V |
|---|---|---|---|---|---|
| **Environmental Disaster** | | | | | |
| Gender | | | | | |
| • Male | 331 (89.9) [0.864,0.928] | 16 (4.3) [0.025,0.069] | 21 (5.7) [0.035,0.085] | 0.92 | 0.01 |
| • Female | 292 (89.3) [0.854,0.924] | 14 (4.3) [0.023,0.070] | 21 (6.4) [0.040,0.096] | | |
| **Earthquake** | | | | | |
| Gender | | | | | |
| • Male | 291 (79.1) [0.745,0.831] | 34 (9.2) [0.064, 0.126] | 43 (11.7) [0.085,0.154] | 0.00 | 0.150 |
| • Female | 216 (66.1) [0.606,0.711] | 42 (12.8) [0.094,0.169] | 69 (21.1) [0.168,0.259] | | |
| **Outbreak of Influenza** | | | | | |
| Gender | | | | | |
| • Male | 295 (80.2) [0.757,0.841] | 34 (9.2) [0.064, 0.126] | 39 (10.6) [0.076,0.142] | 0.005 | 0.124 |
| • Female | 227 (69.4) [0.641,0.743] | 48 (14.7) [0.110,0.189] | 52 (15.9) [0.121,0.203] | | |
| **Outbreak of SARS** | | | | | |
| Gender | | | | | |
| • Male | 197 (53.5) [0.482,0.587] | 65 (17.7) [0.139,0.219] | 106 (28.8) [0.242,0.337] | 0.45 | 0.047 |
| • Female | 166 (50.8) [0.452,0.563] | 70 (21.4) [0.170,0.262] | 91 (27.8) [0.230,0.330] | | |
| **Mass Casualty Incident** | | | | | |
| Gender | | | | | |
| • Male | 326 (88.6) [0.848,0.916] | 17 (4.6) [0.027,0.072] | 25 (6.8) [0.044,0.098] | 0.000 | 0.163 |
| • Female | 250 (76.5) [0.714,0.809] | 26 (8.0) [0.052,0.114] | 51 (15.6) [0.118,0.199] | | |
| **Chemical Spillage** | | | | | |
| Gender | | | | | |
| • Male | 270 (73.4) [0.685,0.778] | 36 (9.8) [0.069,0.132] | 62 (16.8) [0.131,0.210] | 0.768 | 0.028 |
| • Female | 233 (71.3) [0.660,0.761] | 37 (11.3) [0.080,0.152] | 57 (17.4) [0.134,0.219] | | |
| **Conventional War** | | | | | |
| Gender | | | | | |
| • Male | 347 (94.3) [0.914,0.964] | 12 (3.3) [0.017,0.056] | 9 (2.4) [0.011,0.045] | 0.84 | 0.02 |
| • Female | 305 (93.3) [0.899,0.957] | 13 (4.0) [0.021,0.067] | 9 (2.8) [0.012,0.051] | | |
| **Nuclear War** | | | | | |
| Gender | | | | | |
| • Male | 185 (50.3) [0.450,0.555] | 64 (17.4) [0.136,0.216] | 119 (32.3) [0.275,0.373] | 0.74 | 0.029 |
| • Female | 169 (51.7) [0.461,0.572] | 55 (16.8) [0.129,0.213] | 219 (31.5) [0.615,0.720] | | |

The assessment of the relationship between the ability to respond to disaster scenarios and gender revealed that except for the war scenarios, (conventional war and responding to victims of nuclear war) where female HCP showed more ability, male HCPs indicated a higher ability to respond to the rest of disaster scenarios as shown in Table 8.

Similar to willingness, analysis of the relationship between years of experience and the ability to respond to different disaster scenarios showed no clear trend as compared to other independent variables. In the scenario of environmental disaster, the HCPs with 0–4 years of working experience indicated significantly higher ability as compared to other experience groups (*p*-value = 0.027, Cramer V = 0.125). Although, the overall relationship between years of experience of HCPs and the ability to respond to SARS outbreak didn't show any significance, yet, if the experience was stratified based on various cadres, the paramedics with experience between 5–10 years showed significantly more ability as compared to the paramedics in other working experience groups (*p*-value = 0.039, Cramer V = 0.179).

**Table 6. Ability of healthcare providers.**

| Scenario | Able<br>n(%) [95%CI] | Not Able<br>n(%) [95%CI] | Not Sure<br>n(%) [95%CI] |
|---|---|---|---|
| **Environmental Disaster** | **511 (73.5) [0.700,0.767]** | **66 (9.5) [0.074,0.119]** | **118 (17.0) [0.142,0.199]** |
| • Doctors | 291 (79.1) [0.745,0.831] | 22 (6.0) [0.037,0.089] | 55 (14.9) [0.114,0.190] |
| • Nurses | 116 (68.2) [0.606,0.751] | 18 (10.6) [0.064,0.162] | 36 (21.2) [0.152,0.280] |
| • Paramedics | 104 (66.2) [0.582,0.735] | 26 (16.6) [0.111,0.233] | 27 (17.2) [0.116,0.240] |
| **Earthquake** | **480 (69.1) [0.207,0.286]** | **61(8.8) [0.098,0.160]** | **154 (22.2) [0.279,0.364]** |
| • Doctors | 258 (70.1) [0.651,0.747] | 29 (7.9) [0.053,0.111] | 81 (22.0) [0.178,0.266] |
| • Nurses | 118 (69.4) [0.618,0.762] | 15 (8.8) [0.050,0.141] | 37 (21.8) [0.158,0.287] |
| • Paramedics | 104 (66.2) [0.582,0.735] | 17 (10.8) [0.064,0.167] | 36 (22.9) [0.166,0.303] |
| **Outbreak of Influenza** | **573 (82.4) [0.794,0.852]** | **44 (6.3) [0.046,0.084]** | **78 (11.2) [0.089,0.138]** |
| • Doctors | 304 (82.6) [0.783,0.863] | 23 (6.2) [0.783,0.863] | 41(11.1) [0.081,0.148] |
| • Nurses | 133 (78.2) [0.712,0.841] | 13 (7.6) [0.041,0.127] | 24 (14.1) [0.092,0.202] |
| • Parmedics | 136 (86.6) [0.802,0.915] | 8 (5.1) [0.022,0.097] | 13 (8.3) [0.044, 0.137] |
| **Outbreak of SARS** | **458 (65.9) [0.622,0.694]** | **169 (24.3) [0.211,0.276]** | **68 (9.8) [0.076,0.122]** |
| • Doctors | 248 (67.5) [0.623,0.721] | 83 (22.4) [0.183,0.271] | 37 (10.1) [0.071,0.135] |
| • Nurses | 107 (62.9) [0.552,0.702] | 48 (28.2) [0.216,0.356] | 15 (8.8) [0.050,0.141] |
| • Paramedics | 103 (65.6) [0.576,0.729] | 38 (24.2) [0.177,0.316] | 16 (10.2) [0.059,0.1602] |
| **Mass Casualty Incident** | **633 (91.1%) [0.887,0.930]** | **24 (3.5) [0.022,0.050]** | **38 (5.5) [0.039,0.074]** |
| • Doctors | 335 (91.0) [0.876,0.937] | 14 (3.8) [0.020,0.063] | 19 (5.2) [0.031,0.079] |
| • Nurses | 155 (91.2) [0.858,0.949] | 6 (3.5) [0.013,0.075] | 9 (5.3) [0.024,0.098] |
| • Paramedics | 143 (91.1) [0.854,0.950] | 4 (2.5) [0.007,0.063] | 10 (6.4) [0.031,0.114] |
| **Chemical Spillage** | **475 (68.3) [0.647,0.717]** | **92 (13.2) [0.108,0.159]** | **128 (18.4) [0.156,0.215]** |
| • Doctors | 257 (69.1) [0.648,0.744] | 52 (14.8) [0.107,0.181] | 59 (16.0) [0.124,0.201] |
| • Nurses | 111 (65.3) [0.576,0.724] | 22 (12.9) [0.082,0.189] | 37 (21.8) [0.158,0.287] |
| • Paramedics | 107 (68.2) [0.602,0.753] | 18 (11.4) [0.069,0.175] | 32 (20.4) [0.143,0.275] |
| **Conventional War** | **670 (96.4) [0.947,0.976]** | **7 (1.0) [0.004,0.020]** | **18 (2.6) [0.015,0.040]** |
| • Doctors | 355 (96.5) [0.940,0.981] | 2 (0.5) [0.0007,0.0195] | 11 (3.0) [0.015,0.052] |
| • Nurses | 163 (95.9) [0.917,0.983] | 5 (2.9) [0.009,0.067] | 2 (1.2) [0.001,0.041] |
| • Paramedics | 152 (96.8) [0.927,0.989] | 0 (0) [0.0000] | 5 (3.2) [0.010,0.072] |
| **Nuclear War** | **376 (54.1) [0.503,0.578]** | **86 (12.4) [0.100,0.150]** | **233 (33.5) [0.300,0.371]** |
| • Doctors | 215 (58.4) [0.532,0.635] | 44 (12.0) [0.088,0.157] | 109 (29.6) [0.250,0.345] |
| • Nurses | 83 (48.8) [0.410,0.565] | 18 (10.6) [0.064,0.162] | 69 (40.6) [0.331,0.483] |
| • Paramedics | 78 (49.7) [0.416,0.577] | 24 (15.3) [0.100,0.218] | 55 (35.0) [0.276,0.430] |

**SARS:** Severe Acute Respiratory Syndrome

The data analysis suggests that as compared to willingness there were fewer significant relationships between dependent and independent variables in the ability of surveyed HCPs to respond to the given disaster scenarios indicating, that fewer variables influence the HCPs' ability to respond to the disasters. The survey revealed that 63 HCPs showed their ability to respond to all types of disaster scenarios as compared to 59 HCPs who indicated their willingness to respond to all types of given disaster scenarios.

The highest both willingness and ability to respond to a disaster by HCPs was shown in responding to conventional war 648 (94.59%) [95% CI 0.911,0.949]. The lowest willingness and ability were indicated in the scenario of nuclear war in which only 314 (45.83%) [95% CI 0.414,0.489] of HCPs showed both willingness and ability to respond (Fig 1).

**Table 7. Childcare obligation and ability to respond to disasters.**

| Scenario | Able n(%) [95%CI] | Not Able n(%) [95%CI] | Not Sure n(%) [95%CI] | p | Cramer V |
|---|---|---|---|---|---|
| **Environmental Disaster** | | | | | |
| Obligation of Childcare | | | | | |
| • Yes | 95 (64.2) [0.559,0.718] | 16 (10.8) [0.063,0.169] | 37 (25.0) [0.182,0.327] | 0.008 | 0.118 |
| • No | 416 (76.1) [0.722,0.795] | 50 (9.1) [0.068,0.118] | 81 (14.8) [0.119,0.180] | | |
| **Earthquake** | | | | | |
| Obligation of Childcare | | | | | |
| • Yes | 92 (62.2) [0.538,0.700] | 14 (9.5) [0.052,0.153] | 42 (28.4) [0.212,0.363] | 0.096 | 0.082 |
| • No | 388 (70.9) [0.669,0.747] | 47 (8.6) [0.063,0.112] | 112 (20.5) [0.171,0.241] | | |
| **Outbreak of Influenza** | | | | | |
| Obligation of Childcare | | | | | |
| • Yes | 112 (75.7) [0.679,0.823] | 14 (9.5) [0.052,0.153] | 22 (14.9) [0.095,0.216] | 0.046 | 0.094 |
| • No | 461 (84.3) [0.809,0.872] | 30 (5.5) [0.037,0.077] | 56 (10.2) [0.078,0.130] | | |
| **Outbreak of SARS** | | | | | |
| Obligation of Childcare | | | | | |
| • Yes | 90 (60.8) [0.524,0.687] | 46 (31.1) [0.237,0.392] | 12 (8.1) [0.042,0.137] | 0.090 | 0.083 |
| • No | 368 (67.3) [0.631,0.712] | 123 (22.5) [0.190,0.262] | 56 (10.2) [0.078,0.130] | | |
| **Mass Casualty Incident** | | | | | |
| Obligation of Childcare | | | | | |
| • Yes | 139 (93.9) [0.887,0.971] | 2 (1.4) [0.001,0.048] | 7 (4.7) [0.019,0.095] | 0.25 | 0.063 |
| • No | 494 (90.3) [0.875,0.926] | 22 (4.0) [0.025,0.060] | 31 (5.7) [0.038,0.079] | | |
| **Chemical Spillage** | | | | | |
| Obligation of Childcare | | | | | |
| • Yes | 98 (66.2) [0.579,0.737] | 22 (14.9) [0.095,0.216] | 28 (18.9) [0.129,0.261] | 0.768 | .028 |
| • No | 377 (68.9) [0.648,0.727] | 70 (12.8) [0.101,0.158] | 100 (18.3) [0.151,0.217] | | |
| **Conventional War** | | | | | |
| Obligation of Childcare | | | | | |
| • Yes | 142 (95.9) [0.913,0.985] | 3 (2.0) [0.004,0.058] | 3 (2.0) [0.004,0.058] | 0.337 | 0.056 |
| • No | 528 (96.5) [0.946,0.979] | 4 (0.7) [0.002,0.018] | 15 (2.7) [0.015,0.044] | | |
| **Nuclear War** | | | | | |
| Obligation of Childcare | | | | | |
| • Yes | 64 (43.2) [0.351,0.516] | 14 (9.5) [0.052,0.153] | 70 (47.3) [0.390,0.556] | 0.000 | 0.152 |
| • No | 376 (54.1) [0.646,0.726] | 86 (12.4) [0.127,0.190] | 233 (33.5) [0.384,0.468] | | |

Every disaster scenario had a different frequency of barriers to the ability to respond but one theme that emerged consistently was a lack of capability (expertise/knowledge/skills) among the HCPs which reduced their ability to respond. 291 (41.9%) [95% CI 0.381,0.456] of HCPs cited lack of capability as a barrier to their ability to respond to disasters as shown in the Table 9.

## Discussion

The health system and in particular tertiary health care system faces a shortage of human resources. These shortages compromise the systems' functionality, sustainability, and ability to respond to disasters [24]. Even if the health system has adequate resources, certain disasters or public health emergencies can easily overwhelm the preexisting capacities [25].

Recently, hospitals have felt the need for preparedness assessment encompassing all the elements and plans that predict how many beds are available, what number of HCPs are required,

**Table 8. Gender and ability to respond to disasters.**

| Scenario | Able n(%) [95%CI] | Not Able n(%) [95%CI] | Not Sure n(%) [95%CI] | P | Cramer V |
|---|---|---|---|---|---|
| **Environmental Disaster** | | | | | |
| Gender | | | | | |
| • Male | 274 (74.5) [0.696,0.788] | 37 (10.1) [0.071,0.135] | 57 (15.5) [0.119,0.1960] | 0.50 | 0.044 |
| • Female | 237 (72.5) [0.672,0.772] | 29 (8.9) [0.060,0.124] | 61 (18.7) [0.145,0.233] | | |
| **Earthquake** | | | | | |
| Gender | | | | | |
| • Male | 259 (70.4) [0.654,0.750] | 34 (9.2) [0.064,0.126] | 75 (20.4) [0.163,0.248] | 0.47 | 0.046 |
| • Female | 221 (67.6) [0.622,0.726] | 27 (8.3) [0.055,0.117] | 79 (24.2) [0.196,0.291] | | |
| **Outbreak of Influenza** | | | | | |
| Gender | | | | | |
| • Male | 313 (85.1) [0.809,0.885] | 21 (5.7) [0.035,0.0859] | 34 (9.2) [0.064,0.126] | 0.144 | 0.075 |
| • Female | 260 (79.5) [0.747,0.837] | 23 (7.0) [0.045,0.103] | 44 (3.5).[0.099,0.176] | | |
| **Outbreak of SARS** | | | | | |
| Gender | | | | | |
| • Male | 255 (69.3) [0.643,0.739] | 78 (21.2) [0.171,0.257] | 35 (9.5) [0.067,0.129] | 0.102 | 0.0814 |
| • Female | 203 (62.1) [0.565,0.673] | 91 (27.8) [0.230,0.330] | 33 (10.1) [0.070,0.138] | | |
| **Mass Casualty Incident** | | | | | |
| Gender | | | | | |
| • Male | 339 (92.1) [0.888,0.946] | 14 (3.8) [0.020,0.063] | 15 (4.1) [0.023,0.066] | 0.208 | 0.061 |
| • Female | 294 (89.9) [0.861,0.929] | 10 (3.1) [0.014,0.055] | 23 (7.0) [0.045,0.103] | | |
| **Chemical Spillage** | | | | | |
| Gender | | | | | |
| • Male | 262 (71.2) [0.662,0.757] | 45 (12.2) [0.090,0.160] | 61 (16.6) [0.129,0.207] | 0.226 | 0.065 |
| • Female | 213 (65.1) [0.597,0.703] | 47 (14.4) [0.107,0.186] | 67 (20.5) [0.162,0.252] | | |
| **Conventional War** | | | | | |
| Gender | | | | | |
| • Male | 354 (96.2) [0.937,0.979] | 1 (0.3) [0.0001,0.0150] | 13 (3.5) [0.018,0.059] | 0.032 | 0.100 |
| • Female | 316 (96.6) [0.940,0.983] | 6 (1.8) [0.006,0.039] | 5 (1.5) [0.005,0.035] | | |
| **Nuclear War** | | | | | |
| Gender | | | | | |
| • Male | 197 (53.4) [0.482,0.587] | 54 (14.7) [0.112,0.187] | 117 (31.8) [0.270,0.368] | 0.129 | 0.77 |
| • Female | 179 (54.7) [0.491,0.602] | 32 (9.8) [0.067,0.135] | 116 (35.5) [0.302,0.409] | | |

and more importantly how many of them will be willing and available for the response. Given the importance of HCPs' role in responding to disasters, it is pertinent to estimate their willingness and ability and understand the factors that influence their decision to report to work for effective disaster response.

This survey which enrolled 695 HCPs including doctors, nurses and paramedics across the tertiary healthcare system of KP province was conducted to determine the ability and willingness of HCPs to respond to different disaster scenarios.

The results of this survey showed that not all HCPs will be willing or able to respond during a disaster. For different disaster scenarios, between 6% and 47% of HCP indicated their unwillingness, and between 3% & 41% indicated that they were unable to respond to the given disaster scenarios. No single disaster scenario generated either 100% willingness or ability by the HCPs. These results are not different from the results of other studies that have been conducted to assess either willingness or ability. No previously conducted study has shown 100%

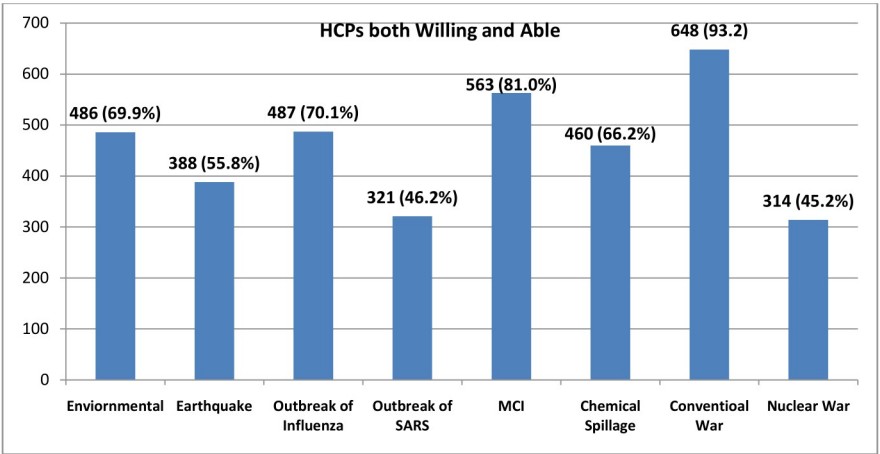

**Fig 1. Healthcare providers both willing and able to respond to disasters.**

willingness or ability in disaster scenarios. Literature has shown that between 10%-50% of HCPs will either be unwilling or unable to respond to disasters [15, 21].

The assessment of willingness revealed that for disaster scenarios of nuclear war, SARS outbreak, and chemical spillage, the willingness of the HCPS was considerably less than that of the other disaster scenarios. This finding reinforces the results emerging from previous researches in this field which showed that Chemical, Biological, Radiological, and Nuclear (CBRN) disasters incite lower willingness as compared to other disasters [11, 15, 21]. These findings can be explained by Kasperson's framework of social amplification of risk which highlights that characteristics of risk (how dreaded and unknown) and how it is interoperated and communicated by social actors influence one's decision [26]. The greater the perceived dread and lesser familiarity with the event, the greater will be the risk perception and fear generation which will result in aversion behavior and low willingness to respond [27].

The highest both willingness (93.8%) and ability (96.4%) reported by HCPs was for reporting to the duty during a conventional war. The reason for this high willingness and ability can be many folds. Probably the most important is the sense of obligation to respond to a national cause and a sense of patriotism [28]. Nationalism or obligation of patriotic duty has been shown to decrease fear and aversion during the war [29]. Secondly, the conventional war involves patient care with which most of the HCPs of the KP province already have experience. They have been regularly managing victims of war since the early 80s with the start of the

**Table 9. Barriers to ability to respond to disasters.**

| Healthcare Providers | Lack of Capability (Expertise/Knowledge/skills) n(%) [95%CI] | Childcare Obligation n(%) [95%CI] | Elderly Care Obligation n(%) [95%CI] | Transportation n(%) [95%CI] | Medical Condition n(%) [95%CI] |
|---|---|---|---|---|---|
| Doctors | 172 (46.7) [0.415,0.519] | 59 (16.0) [0.124,0.201] | 75 (20.4) [0.163,0.248] | 44 (12.0) [0.088,0.157] | 18 (4.9) [0.029,0.076] |
| Nurses | 47 (27.6) [0.210,0.350] | 50 (29.4) [0.226,0.368] | 16 (9.4) [0.054,0.148] | 34 (20.0) [0.142,0.268] | 23 (13.5) [0.087,0.196] |
| Paramedics | 72 (45.9) [0.378,0.539] | 13 (8.3) [0.044,0.137] | 26 (16.6) [0.111,0.233] | 16 (10.2) [0.059,0.160] | 30 (19.1) [0.132,0.261] |
| **Total** | **291 (41.9) [0.381,0.456]** | **122 (17.6) [0.148,0.205]** | **117 (16.8) [0.141,0.198]** | **94 (13.5) [0.110,0.163]** | **71 (10.2) 0.080,0.127]** |

Afghan war and later on as a result of the war on terror and its aftermaths. The scenarios of conventional war and nuclear war have not been explored in previously conducted studies. Pakistan, with its current geopolitical situation and context of the recent escalation in tensions with its neighbor India, is constantly in danger of indulging in a conventional and or nuclear war. Thus, making these scenarios relevant to Pakistan and KP province.

The study results showed that gender had a significant correlation with the willingness to respond to certain disaster scenarios (p = 0.000 for earthquake, p = 0.005 for Influenza outbreak, and p = 0.000 for MCI). Male HCPs showed a higher willingness to respond to 7 out of 8 disaster scenarios. This finding is consistent with the evidence which suggests that across a wide range of real world scenarios, men engage in risky behavior more than females [30] as females' perception of hazards is more severe than males [31]. A study by Mather showed that this difference in risky behavior is further amplified under stress and males take more risks under stressful situations like disasters [32]. However, when the association of ability was correlated with gender, a significant association was observed only in one given scenario which was responding to conventional war (p = 0.032). Interestingly, females (96.6%) significantly showed more ability as compared to male HCPs (96.2%). This difference in ability is mostly attributed to the nurses which were more than 92% female and overall 95.9% of nurses indicated the ability to respond to conventional war. In no other scenario apart from responding to victims of conventional war did the results show a significant relationship between ability and gender.

The survey highlighted that HCPs with child care obligations showed a significantly lower willingness to respond to a disaster as compared to HCPs with no obligation in all scenarios. However, this association was significant (p<0.05) in 4 out of 8 disaster scenarios (earthquake, MCI, an outbreak of Influenza, and SARS). Similarly, childcare obligation was significantly associated with a lower ability to respond to disaster in 3 out of 8 scenarios i.e. environmental disaster (p = 0.008), an outbreak of Influenza (p = 0.046), and nuclear war (p = 0.000). This relationship between childcare obligation and willingness and ability has also been explored in multiple researches and the generated evidence suggests that the health professionals with childcare obligation have a lower willingness and ability to respond to duty during disasters [15, 15, 33, 34].

The biggest barrier identified in this study to the willingness to respond to a disaster was a concern for the family which was indicated by 51.2% of HCPs followed by concern for self in 28.3% of HCPs. Multiple studies have identified both concern/fear for family and concern/fear for self as the biggest barriers to the willingness to respond to disasters [12, 15, 21, 22, 35–37]. The HCPs are likely to be more concerned for themselves and their families than the average citizen. This is because HCPs have a better understanding of the risk associated with a particular disaster [21]. The concern for family and concern for self which are the major barriers to the willingness can also be explained by Killian's theory of "Role Conflict" which suggests that HCPs will face a clash and dilemma between their role in hospitals and their role and obligation towards the family in any disaster situation and the HCPs will incline to the responsibility towards family [38]. The role conflict theory's assumptions are supported by studies that show that for HCPs, family comes first [12, 22, 35–37].

The survey results suggest that the HCPs were more able than willing in 5 out of 8 scenarios which were: i) outbreak of Influenza ii) outbreak of SARS iii) MCI iv) Conventional war and v) Nuclear war. Similar findings have been reported by multiple studies indicating that HCPs are more likely to indicate ability than the willingness to respond to disasters [15, 21, 22]. The findings of the survey indicate that disaster scenarios that are associated with high perceived risk results in an indication of low willingness as compared to ability. The ability to respond depends on factors that are fairly different from the factors that affect willingness in most

cases. The ability is not influenced by the risk perception as explained by the Kasperson framework earlier [26].

The results also showed that disaster scenarios with complex management such as responding to the outbreak of SARS, chemical spillage, and nuclear war provoked relatively lesser ability by the HCPs as compared to other disaster scenarios. These findings further strengthen the evidence generated by the previous researches which shows that the HCPs responding to complex and rare disasters indicate a relatively lower ability as compared to other disasters [15, 21, 22].

The biggest barrier to the ability reported by the HCPs was a lack of capability (knowledge, expertise, and skills) to respond to the disaster. Although, most of the HCPs understand and realize the importance of their role in response to a disaster, yet certain assertions like perceived lack of training, perceived lack of capabilities, and lack of appropriate tools and directions, diminish their ability to respond to a disaster [39].

This survey highlighted that only 63 (9.06%) HCPs indicated their ability to every disaster scenario presented to them. Similarly, 62 (8.92%) of HCPs indicated that they are willing to report/respond to all the given disaster scenarios. Very few researches have explored this phenomenon. A study conducted by Jeffory demonstrated that the percentage of HCPs who are both willing and able is substantially low to what general ability and willingness are indicated for a single event [40].

## Recommendations

The willingness and ability of the HCPs to perform their duty during disasters can be affected by many factors. The greater the perceived dread and lesser familiarity with the event, such as disaster scenarios of nuclear war, SARS outbreak, and chemical spillage, greater will be the risk perception and fear inducing which will evidently result in aversion behavior and low willingness to respond. The health system can work on the concerns of HCPs and some of them can be susceptible to an amendment by certain interventions. The most important interventions that can reduce concerns are education, providing HCPs with personal protective equipment, provision of vaccines to HCPS and their families. For childcare obligation, which has been highlighted as a major factor, hospitals can plan and arrange a safe and secure space for the children of HCPs responding to various disaster.

Another factor that was not studied in the survey are the mental health needs of the HCPs in disaster scenarios. Literature suggests that a calamity can emotionally traumatize people affected by it and may lead to Post Traumatic Stress Disorder (PTSD) [41]. Furthermore, if someone experiences repeated disasters, the mental health consequences can multiply [42]. The HCPs of KP province have over the past decade experienced repeated manmade disasters as well as natural hazards. These disasters must have taken a toll on their mental health. Their willingness and ability may also be linked to their mental health. Addressing their mental health needs post disaster response may improve their willingness and ability to respond to future disasters.

Keeping in view the given facts, the hospitals and mangers need to have separate human resources management plans in designated disaster plans as different disasters will generate different reporting and absenteeism rates from HCPs. Hospitals need to pay attention to trainings, provide supportive environment, address mental health needs and reassurance of the HCPs especially for the rare disaster events to improve willingness and ability to report and respond which will improve surge capacity.

## Limitations

There are certain limitations to our study. This research used Perception model to assess the willingness and ability of HCPs to respond to various disaster scenarios. There are certain

limitations to this technique. The foremost is the written scenarios that might not explain the potential threat of the disaster to the fullest which forms the basis for the response. The scenarios might not be able to imitate the actual events and are unable to inform the true extent of threat one might face and this leads to normalcy bias as respondents are unable to interpret risk properly and may misinterpret high risk situation as not threatening with low or minimal risk situation. However, despite these limitations and keeping in view the HCPs experience with various disasters in the KP province over the years, the Perception model was deemed most suitable to assess the willingness and ability of the HCPs to respond to the various disaster scenarios.

The survey tool used to assess willingness and ability been used in various studies. We used a slightly modified version with addition of few additional disaster scenarios. Although it is a valid tool, its reliability/consistency of the survey tool was not measured.

Information about non responders' willingness and ability is not known, hence, responder bias must be considered when interoperating the results. Importantly, we might not know how the HCPs will actually respond to the disaster, whether they will respond as they indicated to different disaster scenarios or will they act differently. This can only be know in an actual disaster scenario with real threat perception shaping their willingness and ability to respond

## Conclusion

During disaster response, the major challenge the hospital can face is the availability of adequate human resources to respond to the sudden surge of the patients. It is always challenging to anticipate the type of disaster and the reporting behavior of the HCPs as a result of that disaster.

This study showed that not all the HCPs on the tertiary care hospital's roster will be available during disaster response. Their numbers may differ during different disaster scenarios. This difference stems from various factors influencing the decision of HCPs to report during disasters. The biggest barriers to the willingness and ability of the HCPs are childcare obligation and lack of capability respectively.

This unwillingness and inability of HCPs to respond to disasters greatly limit the tertiary health care system's potential surge capacity to effectively manage the disasters. Major hospitals and health system, in general, need to estimate the reporting behavior of HCPs and identify the barriers to reporting to plan for effective disaster response. The majority of these barriers are amendable and addressing them with appropriate interventions will improve the availability of human resources which will in turn improve the surge capacity of the hospital and health system in general.

This study provides an opportunity for the tertiary healthcare system of not only Khyber Pakhtunkhwa but other similar health systems within the country or in other countries to build on the findings and amend and include mitigation plans to address the concerns and barriers that might hinder the HCPs' willingness or ability to report/respond to the disasters and improve the overall disaster response capabilities.

## Supporting information

**S1 File. Willingness & ability questionnaire.**
(DOCX)

## Author Contributions

**Conceptualization:** Muhammad Zeeshan Haroon, Inayat Hussain Thaver.

**Data curation:** Muhammad Zeeshan Haroon.

**Formal analysis:** Muhammad Zeeshan Haroon, Muhammad Imran Marwat.

**Investigation:** Muhammad Zeeshan Haroon.

**Methodology:** Muhammad Zeeshan Haroon.

**Project administration:** Muhammad Zeeshan Haroon.

**Supervision:** Inayat Hussain Thaver.

**Writing – original draft:** Muhammad Zeeshan Haroon, Muhammad Imran Marwat.

**Writing – review & editing:** Muhammad Zeeshan Haroon, Inayat Hussain Thaver.

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
