## [Decision Letter · Decision Letter 0]

31 Jan 2022

PONE-D-21-11986Are the healthcare providers willing and able to respond to disasters: An assessment of tertiary health care system of Khyber PakhtunkhwaPLOS ONE

Dear Dr. Haroon,

Thank you for submitting your manuscript to PLOS ONE. After careful consideration, we feel that it has merit but does not fully meet PLOS ONE’s publication criteria as it currently stands. Therefore, we invite you to submit a revised version of the manuscript that addresses the points raised during the review process.

The reviewers have raised a number of concerns that need attention. They request additional information on methodological aspects of the study (such as the technical statistical details ), as well as the overall reporting of the results section. Furthermore the manuscript requires extensive editing for English grammar and usage, one of the publication criteria at PLOS ONE (https://journals.plos.org/plosone/s/criteria-for-publication#loc-5 )is that articles must be presented in an intelligible fashion and written in clear, correct, and unambiguous English. Please note that PLOS ONE cannot provide copyediting for manuscripts 

Could you please revise the manuscript to carefully address the concerns raised?

We look forward to receiving your revised manuscript.

Kind regards,

Lucinda Shen, MSc

Staff Editor

PLOS ONE

Journal Requirements:

2. PLOS ONE does not copy edit accepted manuscripts (https://journals.plos.org/plosone/s/criteria-for-publication#loc-5). To that effect, please ensure that your submission is free of typos and grammatical errors.

Reviewers' comments:

Reviewer's Responses to Questions

**Comments to the Author**

1. Is the manuscript technically sound, and do the data support the conclusions?

Reviewer #1: Partly

Reviewer #2: Partly

2. Has the statistical analysis been performed appropriately and rigorously? 

Reviewer #1: No

Reviewer #2: Yes

3. Have the authors made all data underlying the findings in their manuscript fully available?

Reviewer #1: Yes

Reviewer #2: Yes

4. Is the manuscript presented in an intelligible fashion and written in standard English?

Reviewer #1: Yes

Reviewer #2: No

5. Review Comments to the Author

Reviewer #1: You need to explain in the Methods section the statistical analysis and the statistical tests you have made and also because you are using a sample you will need to include in the Results section teh confidence intervals or the parameters

Reviewer #2: Thank you for the opportunity to review this manuscript. The considerable work that has gone into the preparation and interesting findings reported are appreciated but the manuscript as a whole lacks sufficient organization and cohesive explication of major findings and implications.

Some specific comments:

Introduction- the narrative is not well-focused, description of Khyber Pakhtunkhwa (KP) province cursory, local demographics are not described

Methodology- while using a prior tool that has been adapted with additional scenarios, this survey does not differentiate or provide further details about what specific contexts within the scenario willingness to respond applies.

Results- good sample size and excellent response rate

Discussion- reasonable discussion of findings, but implications of findings not addressed; no limitations of study described.

Tables- some typos "mass causality"

6. PLOS authors have the option to publish the peer review history of their article (what does this mean?). If published, this will include your full peer review and any attached files.

Reviewer #1: No

Reviewer #2: No

---

## [Author Response · Author response to Decision Letter 0]

9 May 2022

Response To Reviewers

Journal Requirement 

Comment 1:

In your Data Availability statement, you have not specified where the minimal data set underlying the results described in your manuscript can be found

Reply:

This work is part of my PhD thesis. As per the directions of the institute, we are not suppose to make publicly available our data set. The reason behind is that it may be used by someone else in their publication and when our thesis plagiarism is checked, it may appear that a copied data is used in the thesis

The data is available with author and if the journal whishes to have the data, SPSS file can be sent to journal for verification of results written in the paper. Additionally all the tables from which results are extracted are already mentioned in the article

Comment 2:

Please include your full ethics statement in the ‘Methods’ section of your manuscript file. In your statement, please include the full name of the IRB or ethics committee who approved or waived your study, as well as whether or not you obtained informed written or verbal consent.

Reply:

Details added in the methodology section please see line 116-120. Regarding consent, details are given in the line 122-123

Reviewer Comments

Reviewer #1:

You need to explain in the Methods section the statistical analysis and the statistical tests you have made and also because you are using a sample you will need to include in the Results section the confidence intervals or the parameters

Reply:

Statistical analysis and tests have now been described at the end of the methodology section. Please refer to line 125-128

All the results and tables have now been updated and 95% confidence intervals have been added.

Reviewer #2: Thank you for the opportunity to review this manuscript. The considerable work that has gone into the preparation and interesting findings reported are appreciated but the manuscript as a whole lacks sufficient organization and cohesive explication of major findings and implications.

Reply:

I apologies but I have tried to organize it to best of my ability. Can the reviewer please comment on section which needs improvement? And any suggestion will be highly appreciated

Specific comments:

Introduction- the narrative is not well-focused, description of Khyber Pakhtunkhwa (KP) province cursory, local demographics are not described

Reply:

Information related to KP health system has been added in addition to information related to disasters and affects that were already mentioned. Please refer to lines 47-59

Methodology- while using a prior tool that has been adapted with additional scenarios, this survey does not differentiate or provide further details about what specific contexts within the scenario willingness to respond applies

Reply:

This survey builds upon the prior tool and added additional disaster scenarios which have not been studied by the previous researches. The additional disaster scenarios are very relevant to the KP province and Pakistan and this survey has assessed the willingness and ability of HCPs for these scenarios. Furthermore this study has also explored the proportion of HCPs who are both willing and able to respond to different disaster scenarios

Discussion- reasonable discussion of findings, but implications of findings not addressed; no limitations of study described. 

Reply:

Implications have been discussed in the last couple of paragraphs of Conclusion (please refer to lines 421-432) 

Limitation section added please refer to lines 433-443

Tables- some typos "mass causality

Reply:

Manuscript was reviewed and spelling and grammatical mistakes were corrected

---

## [Decision Letter · Decision Letter 1]

13 Oct 2022

PONE-D-21-11986R1Are the healthcare providers willing and able to respond to disasters: An assessment of tertiary health care system of Khyber PakhtunkhwaPLOS ONE

Dear Dr. Haroon,

Thank you for submitting your manuscript to PLOS ONE. After careful consideration, we feel that it has merit but does not fully meet PLOS ONE’s publication criteria as it currently stands. Therefore, we invite you to submit a revised version of the manuscript that addresses the points raised during the review process.

We look forward to receiving your revised manuscript.

Kind regards,

Abdallah Y Naser, PhD

Academic Editor

PLOS ONE

Additional Editor Comments:

Dear authors

Based on the authors revision. I would recommend you address the following comments before resubmission:

- Add the ethical approval number in the methodology section.

- Add the name of the software used for statistics and more details regarding the analysis. Chi-square test is not used to measure association, please mention why did you used Cramer V statistical tests specifically. Why the authors did not do some regression analysis. The data could be presented better than in the format of proportion only. You could have done some score scale for each outcome of interest and then stratify participants demographic based on this score, then you can do regression analysis to identify predictors. Refer to this article as a reference to understand what i mean (https://pubmed.ncbi.nlm.nih.gov/32578943/)

- The manuscript required professional English language editing to enhance the flow. Please provide us with certificate of accomplishing this upon re-submission.

- Add further details about sample size calculation including sample size of your population and what is the minimum required sample size (include the formula used).

- Describe exactly how did you apply your recruitment procedure (stratified random sampling).

- Further details are needed regarding questionnaire tool and their assessment scale (format of questions).

- Please address all comments and amend the discussion accordingly.

Best regards

Reviewers' comments:

Reviewer's Responses to Questions

**Comments to the Author**

1. If the authors have adequately addressed your comments raised in a previous round of review and you feel that this manuscript is now acceptable for publication, you may indicate that here to bypass the “Comments to the Author” section, enter your conflict of interest statement in the “Confidential to Editor” section, and submit your "Accept" recommendation.

Reviewer #3: (No Response)

Reviewer #4: (No Response)

2. Is the manuscript technically sound, and do the data support the conclusions?

Reviewer #3: Partly

Reviewer #4: Yes

3. Has the statistical analysis been performed appropriately and rigorously? 

Reviewer #3: Yes

Reviewer #4: Yes

4. Have the authors made all data underlying the findings in their manuscript fully available?

Reviewer #3: Yes

Reviewer #4: Yes

5. Is the manuscript presented in an intelligible fashion and written in standard English?

Reviewer #3: Yes

Reviewer #4: Yes

6. Review Comments to the Author

Reviewer #3: This is a review of the paper entitled "Are the healthcare providers willing and able to respond to disasters: An assessment of tertiary health care system of Khyber Pakhtunkhwa." Thank you for the opportunity to review this manuscript. Unfortunately, I still believe the manuscript lacks sufficient organization and cohesive explication of major findings and implications.

I understand that there has been a lot of work and effort spent on the data collection and analysis since it is based on a dissertation, but it’s not logical to fit all your findings in one academic paper. The current manuscript goes back and forth between such big topics and does not really dig deep into one of them. The paper still lacks focus and does not have a clear message. I don’t need 10 pages of tables and data to know that not everyone in the medical force will be not willing or not able to respond; we already know that. I need to see what this article has to add!

It would be more beneficial to see some more in-depth discussion and clarifications; hopefully, the points raised here will aid the process:

1) Submit a blinded manuscript (no names) next time, because this peer review is a blind process. There is no need for me to see the names of the authors and affiliations in the submittal.

2) There is no need to submit the original report again; we already have it in the system.

3) Don’t use the track changes option in Microsoft Words in the modified manuscript; it makes the document looks messy. Simply highlight only the major changes!

4) Please do not use the 'natural disasters' misnomer! The word ‘Natural’ in a disaster context, suggests the naturalization and normalization of disasters, which is a misleading idea! So much evidence exists pointing out that disasters are not natural; this argument is not new (see O'Keefe et al, 1976: "Taking the 'Naturalness' out of 'Natural Disaster'', as well as more recent work by Kelman or Chmutina et al., in addition to a popular #nonaturaldisasters campaign on Twitter). Please use the term ‘Natural Hazards’ instead, or use ‘disaster’ without the word ‘natural’.

5) In the methodology, I need to know more information about the actual population and how the sample size was determined, and what was the confidence level and confidence interval for the sample. I also want to see the demographics of the original population which the sample was taken from.

6) Your study is based on (Qureshi, K., Gershon, R. R., Sherman, M. F., Straub, T., Gebbie, E., McCollum, M., Erwin, M. J., & Morse, S. S. (2005). Health care workers’ ability and willingness to report to duty during catastrophic disasters. Journal of Urban Health, 82(3), 378–388.) Try to organize your tables and the data you collected in the same way they did! Also, add the limitation in their study to your paper as it should apply here as well. In future studies, try to use a rising scale for your survey, and give the participants more options, for example (not able, able, not sure) 3-point scales are not a good tool to use. Use 5 or 7 points scales.

7) You reflected on the validity at mutable locations in the paper, but you didn’t mention anything about reliability/consistency measures. I understand that the survey tools are previously validated and commonly used, but even for those, measuring reliability/consistency can be helpful. If you have not measured reliability/consistency for your data collected, can you please reflect on that in the limitation as well?

8) In your result section, you don’t need to repeat findings that I can see in the tables in the texts as well.

9) You have too many tables! Reorganize them and try to merge them somehow to make their presentations better. You don’t need to report all the data you collected! If there is no significant finding in one of the tables, don’t include it.

10) Reorganize your discussion and make it in multiple subsections.

11) You need a recommendation section. This is very important! I need to see how I can use your findings! Add the recommendation as a substation at the end of the discussion. Also, the paper does not discuss the social component of the issue or impact on the society in general or what needs to be done to help fix the problem. Integrating mental health in disaster response plans is a very critical topic these days and there are a lot of papers published about it. Adding that to the paper is very important, you can add it to the discussion section, or even make a separate recommendation section. You can start here:

• Cohen, S., & Abukhalaf, A. H. I. (2021). Necessity to Plan and Implement Mental Health Disaster Preparedness and Intervention Plans. Academia Letters. https://doi.org/10.20935/AL3507

• Cohen, S., Abukhalaf, A.H.I. (2021). COVID-19’s Negative Mental Health Impact Goes Well Beyond Standard At-Risk Populations: Action Needs To Be Taken to Combat Long-term Nationwide Emotional Disruption. Academia Letters, Article 3621. https://doi.org/10.20935/AL3621

• Cohen, S., von Meding, D., Abukhalaf, A.H.I. (2021). Successful Pandemic and Disaster Mental Health Preparedness Requires Widespread Community Collaboration. Academia Letters, Article 3987. https://doi.org/10.20935/AL3987

12) The limitation section should be much bigger based on this study and the methodology chosen. Think about the limitations in your study again and expand that section. Move the limitation section to before the conclusion.

13) Add an appendix with the survey question at the end of the report. If the survey was conducted in a different language, provide a translated version of the survey in the English language.

14) If there are no Acknowledgments, remove that section instead of writing (not applicable).

I understand that these comments may seem like a lot to you, but it’s better for you to work on fixing the manuscript and to keep those comments in mind as a good practice for future submittals. Your paper should be well-written, well-organized, well-presented, with a very specific focus and direct and clear massage of implications.

Reviewer #4: The significant effort put into the preparation and the intriguing discoveries presented are great, but the paper has some minor issues and lacks some of the key findings and consequences.

The discussion is an important part of the paper. It should have a reasonable discussion about the results and findings. However, the most important part is the comparison and relating the findings to the similar studies and alternative description of the study. It should be followed by the implication of the study and its solutions to make the study stronger and more reasonable. Finally, the suggestion, which can provide the opportunity for future studies, is worth adding to this part.

1. In the introduction part, there are some statistics about the healthcare facilities in Khyber Pakhtunkhwa, lines 47 to 55, that do not have references. Also, the description of the location and province are not described clearly enough.

2. In methodology, Line 94, explains the perception method based on the quantitative method in the methodology section. Also, describe the statistical tests that you have done for the analysis.

3. Discussion part covers the discussion of results in detail, but there is not a strong implication, comparison, and suggestions.

4. Figure 1 is not clear, create a clear figure with better resolution.

5. Table 11, is not complete. It has messed.

7. PLOS authors have the option to publish the peer review history of their article (what does this mean?). If published, this will include your full peer review and any attached files.

Reviewer #3: **Yes: **Amer Hamad Issa Abukhalaf

Reviewer #4: No

---

## [Author Response · Author response to Decision Letter 1]

27 Jun 2023

Response to Reviewers

Editor Comments

Add the ethical approval number in the methodology section.

Reply: Mentioned and highlighted

Add the name of the software used for statistics and more details regarding the analysis. Chi-square test is not used to measure association, please mention why did you used Cramer V statistical tests specifically. 

Reply: Software name now mentioned in methodology section

Cramer V was used to assess strength of association

Why the authors did not do some regression analysis. The data could be presented better than in the format of proportion only. You could have done some score scale for each outcome of interest and then stratify participants demographic based on this score, then you can do regression analysis to identify predictors. Refer to this article as a reference to understand what i mean (https://pubmed.ncbi.nlm.nih.gov/32578943/)

Reply: Analysis was made as advised by the PhD supervisor

The manuscript required professional English language editing to enhance the flow. Please provide us with certificate of accomplishing this upon re-submission.

Reply: The manuscript has already been edited by the university editor. The university editors don’t issue certificate. If required email copy can be provided

Add further details about sample size calculation including sample size of your population and what is the minimum required sample size (include the formula used).

Reply: Details added

-Describe exactly how did you apply your recruitment procedure (stratified random sampling).

Reply : Details added

Further details are needed regarding questionnaire tool and their assessment scale (format of questions).

Reply: Tool has been uploaded as a separate file

Reviewer #3: 

This is a review of the paper entitled "Are the healthcare providers willing and able to respond to disasters: An assessment of tertiary health care system of Khyber Pakhtunkhwa." Thank you for the opportunity to review this manuscript. Unfortunately, I still believe the manuscript lacks sufficient organization and cohesive explication of major findings and implications.

I understand that there has been a lot of work and effort spent on the data collection and analysis since it is based on a dissertation, but it’s not logical to fit all your findings in one academic paper. The current manuscript goes back and forth between such big topics and does not really dig deep into one of them. The paper still lacks focus and does not have a clear message. I don’t need 10 pages of tables and data to know that not everyone in the medical force will be not willing or not able to respond; we already know that. I need to see what this article has to add!

It would be more beneficial to see some more in-depth discussion and clarifications; hopefully, the points raised here will aid the process:

1) Submit a blinded manuscript (no names) next time, because this peer review is a blind process. There is no need for me to see the names of the authors and affiliations in the submittal.

Reply: Names Removed

2) There is no need to submit the original report again; we already have it in the system.

Reply: Only revised version is shared

3) Don’t use the track changes option in Microsoft Words in the modified manuscript; it makes the document looks messy. Simply highlight only the major changes!

Reply: Corrected as suggested

4) Please do not use the 'natural disasters' misnomer! The word ‘Natural’ in a disaster context, suggests the naturalization and normalization of disasters, which is a misleading idea! So much evidence exists pointing out that disasters are not natural; this argument is not new (see O'Keefe et al, 1976: "Taking the 'Naturalness' out of 'Natural Disaster'', as well as more recent work by Kelman or Chmutina et al., in addition to a popular #nonaturaldisasters campaign on Twitter). Please use the term ‘Natural Hazards’ instead, or use ‘disaster’ without the word ‘natural’.

Reply: Corrected and highlighted

5) In the methodology, I need to know more information about the actual population and how the sample size was determined, and what was the confidence level and confidence interval for the sample. I also want to see the demographics of the original population which the sample was taken from.

Reply: Mentioned and highlighted

6) Your study is based on (Qureshi, K., Gershon, R. R., Sherman, M. F., Straub, T., Gebbie, E., McCollum, M., Erwin, M. J., & Morse, S. S. (2005). Health care workers’ ability and willingness to report to duty during catastrophic disasters. Journal of Urban Health, 82(3), 378–388.) Try to organize your tables and the data you collected in the same way they did! Also, add the limitation in their study to your paper as it should apply here as well. In future studies, try to use a rising scale for your survey, and give the participants more options, for example (not able, able, not sure) 3-point scales are not a good tool to use. Use 5 or 7 points scales.

Reply: Limitations are added as suggested

7) You reflected on the validity at mutable locations in the paper, but you didn’t mention anything about reliability/consistency measures. I understand that the survey tools are previously validated and commonly used, but even for those, measuring reliability/consistency can be helpful. If you have not measured reliability/consistency for your data collected, can you please reflect on that in the limitation as well?

Reply: Mentioned in limitations

8) In your result section, you don’t need to repeat findings that I can see in the tables in the texts as well.

Reply: Results are trimmed and findings which were shown in the table were removed from narration

9) You have too many tables! Reorganize them and try to merge them somehow to make their presentations better. You don’t need to report all the data you collected! If there is no significant finding in one of the tables, don’t include it.

Reply: 2 tables removed

10) Reorganize your discussion and make it in multiple subsections.

Reply Discussion has been trimmed with two clear sub sections. One addressing willingness and other ability

11) You need a recommendation section. This is very important! I need to see how I can use your findings! Add the recommendation as a substation at the end of the discussion. Also, the paper does not discuss the social component of the issue or impact on the society in general or what needs to be done to help fix the problem. Integrating mental health in disaster response plans is a very critical topic these days and there are a lot of papers published about it. Adding that to the paper is very important, you can add it to the discussion section, or even make a separate recommendation section. You can start here:

• Cohen, S., & Abukhalaf, A. H. I. (2021). Necessity to Plan and Implement Mental Health Disaster Preparedness and Intervention Plans. Academia Letters. https://doi.org/10.20935/AL3507

• Cohen, S., Abukhalaf, A.H.I. (2021). COVID-19’s Negative Mental Health Impact Goes Well Beyond Standard At-Risk Populations: Action Needs To Be Taken to Combat Long-term Nationwide Emotional Disruption. Academia Letters, Article 3621. https://doi.org/10.20935/AL3621

• Cohen, S., von Meding, D., Abukhalaf, A.H.I. (2021). Successful Pandemic and Disaster Mental Health Preparedness Requires Widespread Community Collaboration. Academia Letters, Article 3987. https://doi.org/10.20935/AL3987

Reply: Added

12) The limitation section should be much bigger based on this study and the methodology chosen. Think about the limitations in your study again and expand that section. Move the limitation section to before the conclusion.

Reply: Additions done in limitations section

13) Add an appendix with the survey question at the end of the report. If the survey was conducted in a different language, provide a translated version of the survey in the English language.

Reply: Survey Tool uploaded as a separate document

14) If there are no Acknowledgments, remove that section instead of writing (not applicable).

Reply: Removed

I understand that these comments may seem like a lot to you, but it’s better for you to work on fixing the manuscript and to keep those comments in mind as a good practice for future submittals. Your paper should be well-written, well-organized, well-presented, with a very specific focus and direct and clear massage of implications.

Reviewer #4: 

The significant effort put into the preparation and the intriguing discoveries presented are great, but the paper has some minor issues and lacks some of the key findings and consequences.

The discussion is an important part of the paper. It should have a reasonable discussion about the results and findings. However, the most important part is the comparison and relating the findings to the similar studies and alternative description of the study. It should be followed by the implication of the study and its solutions to make the study stronger and more reasonable. Finally, the suggestion, which can provide the opportunity for future studies, is worth adding to this part.

1. In the introduction part, there are some statistics about the healthcare facilities in Khyber Pakhtunkhwa, lines 47 to 55, that do not have references. Also, the description of the location and province are not described clearly enough.

Reply: Reference added

2. In methodology, Line 94, explains the perception method based on the quantitative method in the methodology section. Also, describe the statistical tests that you have done for the analysis.

Reply: Statistical test have been mentioned in methodology

3. Discussion part covers the discussion of results in detail, but there is not a strong implication, comparison, and suggestions.

Reply: Recommendations have been added. Comparison with other studies already discussed in the discussion section

4. Figure 1 is not clear, create a clear figure with better resolution.

Table 11, is not complete. It has messed.

Reply: The word was converted to .tiff format and resolution remains the same. I tried and got same results. Table 11 is now table 9 and its complete

---

## [Editor Report · Decision Letter 2]

19 Oct 2023

Are the healthcare providers willing and able to respond to disasters: An assessment of tertiary health care system of Khyber Pakhtunkhwa

PONE-D-21-11986R2

Dear Dr. Haroon,

We’re pleased to inform you that your manuscript has been judged scientifically suitable for publication and will be formally accepted for publication once it meets all outstanding technical requirements.

Kind regards,

Abdallah Y Naser, PhD

Academic Editor

PLOS ONE
---

## [Editor Report · Acceptance letter]

24 Oct 2023

PONE-D-21-11986R2 

Are the healthcare providers willing and able to respond to disasters: An assessment of tertiary health care system of Khyber Pakhtunkhwa 

Dear Dr. Haroon:

I'm pleased to inform you that your manuscript has been deemed suitable for publication in PLOS ONE. Congratulations! Your manuscript is now with our production department. 

Kind regards, 

on behalf of

Dr. Abdallah Y Naser 

Academic Editor

PLOS ONE